# MULTI-OBJECTIVE-GUIDED DISCRETE FLOW MATCHING FOR CONTROLLABLE BIOLOGICAL SEQUENCE DESIGN

## ABSTRACT

Designing biological sequences that satisfy multiple, often conflicting, functional and biophysical criteria remains a central challenge in biomolecule engineering. While discrete flow matching models have recently shown promise for efficient sampling in high-dimensional sequence spaces, existing approaches address only single objectives or require continuous embeddings that can distort discrete distributions. We present **Multi-Objective-Guided Discrete Flow Matching (MOG-DFM)**, a general framework to steer any pretrained discrete-time flow matching generator toward Pareto-efficient trade-offs across multiple scalar objectives. At each sampling step, MOG-DFM computes a hybrid rank-directional score for candidate transitions and applies an adaptive hypercone filter to enforce consistent multi-objective progression. We also trained two unconditional discrete flow matching models, **PepDFM** for diverse peptide generation and **EnhancerDFM** for functional enhancer DNA generation, as base generation models for MOG-DFM. We demonstrate MOG-DFM's effectiveness in generating peptide binders optimized across five properties (hemolysis, non-fouling, solubility, half-life, and binding affinity), and in designing DNA sequences with specific enhancer classes and DNA shapes. In total, MOG-DFM proves to be a powerful tool for multi-property-guided biomolecule sequence design.

## 1 INTRODUCTION

Designing biological sequences that simultaneously satisfy multiple functional and biophysical criteria is a foundational challenge in modern bioengineering (Naseri & Koffas, 2020). For example, when engineering therapeutic proteins, one must balance high target-binding affinity with low immunogenicity and favorable pharmacokinetics (Tominaga et al., 2024); CRISPR guide RNAs require both high on-target activity and minimal off-target effects (Mohr et al., 2016; Schmidt et al., 2025); and synthetic promoters must achieve strong gene expression while maintaining tissue-specific activation (Artemyev et al., 2024).

Most existing biomolecule-design methods focus on optimizing a single objective in isolation (Zhou et al., 2019; Nehdi et al., 2020). For example, efforts have been made to reduce protein toxicity (Kreiser et al., 2020; Sharma et al., 2022) and neural networks are used to improve protein thermostability (Komp et al., 2025). While these single-objective approaches yield high performance on their target metrics, they often produce sequences with undesirable trade-offs, high-affinity peptides may be insoluble or toxic, and stabilized proteins may lose functional specificity (Bigi et al., 2023; Rinauro et al., 2024). Consequently, a framework for multi-objective guided generation that can balance conflicting requirements is critical to meet the demands of practical biomolecular engineering.

Classical multi-objective optimization (MOO) techniques, such as evolutionary algorithms and Bayesian optimization, have been successfully applied to black-box tuning of molecular libraries (Zitzler & Thiele, 1998; Deb, 2011; Ueno et al., 2016; Frisby & Langmead, 2021). More recently, controllable generative models have been developed to integrate MOO directly into the sampling process (Li et al., 2018; Sousa et al., 2021; Yao et al., 2024). ParetoFlow (Yuan et al., 2024), for instance, leverages continuous-space flow matching to produce Pareto-optimal samples, but operates only in continuous domains. Applying such techniques to discrete sequences typically requires

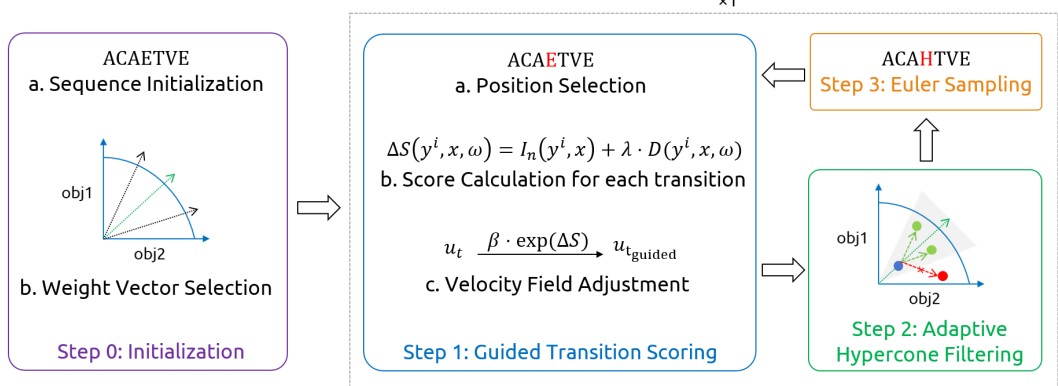

Figure 1: Schematic for MOG-DFM algorithm.

embedding into a continuous manifold, which can distort distributions and complicate property-based guidance (Beliakov & Lim, 2007; Michael et al., 2024).

Discrete flow matching has recently emerged as a powerful paradigm for directly modeling and sampling from complex discrete spaces (Campbell et al., 2024; Gat et al., 2024; Dunn & Koes, 2024). Two primary variants exist: (i) continuous-time simplex methods, which diffuse discrete data through a continuous embedding over the probability simplex (Stark et al., 2024; Davis et al., 2024; Tang et al., 2025a), and (ii) jump-process models that learn time-dependent transition rates for token-level stochastic updates (Campbell et al., 2024; Gat et al., 2024). The latter is particularly well-suited for controllable generation, as it naturally supports reweighting of token transitions based on scalar reward functions.

Recent work has applied these models to single-objective tasks: Nisonoff et al. (2025) introduced rate-based classifier guidance for pretrained samplers, while Tang et al. (2025a) proposed Gumbel-Softmax Flow Matching with straight-through guidance for controllable discrete generation. Yet, to our knowledge, no prior work has extended discrete flow matching to support Pareto-guided generation across multiple objectives.

As such, our key contributions are as follows:

1. **MOG-DFM: Multi-Objective-Guided Discrete Flow Matching**, a general framework that steers pretrained discrete flow matching models toward Pareto-efficient solutions via multi-objective guidance and adaptive hypercone filtering.
2. **Rank-Directional Scoring and Hypercone Filtering** combine rank-normalized local improvement and directional alignment with a user-specified trade-off vector to reweight token-level transition velocities, followed by a dynamic angular filtering mechanism that enforces directional consistency along the Pareto front.
3. **Unconditional Base Models for Biomolecule Generation**; we train two high-quality discrete flow matching models: **PepDFM** for diverse peptide generation and **EnhancerDFM** for functional enhancer DNA generation, demonstrating low loss and biological plausibility.
4. **Multi-Property Sequence Design**; we apply MOG-DFM to two challenging biological generation tasks: (i) therapeutic peptide binder generation with five competing objectives (affinity, solubility, hemolysis, half-life, non-fouling), and (ii) enhancer DNA sequence generation guided by enhancer class and DNA shape.
5. **Superior Multi-Objective Optimization**; MOG-DFM significantly outperforms classical evolutionary and diffusion-based baselines on both peptide and DNA tasks, producing sequences with favorable trade-offs and improved downstream docking, folding, and property scores.

## 2 METHODS

### 2.1 MULTI-OBJECTIVE GUIDED DISCRETE FLOW MATCHING

MOG-DFM (**M**ulti-**O**bjective **G**uided **D**iscrete **F**low **M**atching) operates under the same setting as discrete flow matching described in Section A. Suppose we have a pre-trained discrete flow matching model that defines a CTMC with factorized velocity field $u_t^i(y^i, x)$, which transports probability mass from an initial distribution $p_0$ to the unknown target distribution via mixture path parametrization. In addition, we assume access to $N$ pre-trained scalar score functions $s_n : \mathcal{S} \to \mathbb{R}$, where $n = 1, \ldots, N$, that assign objective scores to any sequence. Our aim is to generate novel sequences $x_1 \in \mathcal{S}$ whose objective vectors $\big(s_1(x_1), s_2(x_1), \ldots, s_N(x_1)\big)$ lie close to the Pareto front (not guaranteed to be Pareto optimal)

$$\text{PF} = \big\{ x \in \mathcal{S} \mid \nexists x' \in \mathcal{S} : s_n(x') \geq s_n(x) \, \forall n, \, \exists m : s_m(x') > s_m(x) \big\}.$$

To achieve this, we will guide the CTMC sampling dynamics of the discrete flow matching model using multi-objective transition scores, steering the generative process toward Pareto-efficient regions of the state space (Figure 1, Pseudo code 1, Proof in Section G).

#### 2.1.1 STEP 0: INITIALIZATION AND WEIGHT VECTOR GENERATION

MOG-DFM begins by initializing the generative process at time $t = 0$ by sampling an initial sequence $x_0$ uniformly from the discrete state space $\mathcal{S} = [K]^d$. To steer the generation towards diverse Pareto-efficient solutions, we introduce a set of weight vectors $\{\omega^k\}_{k=1}^M$ that uniformly cover the $N$-dimensional Pareto Front. Intuitively, each $\omega$ encodes a particular trade-off among the $N$ objectives, so sampling different $\omega$ promotes exploration of distinct regions of the Pareto front. Concretely, we construct these vectors via the Das-Dennis simplex lattice (Das & Dennis, 1998) with $H$ subdivisions, yielding components

$$\omega_i = \frac{k_i}{H}, \quad k_i \in \mathbb{Z}_{\geq 0}, \quad \sum_{i=1}^{N} k_i = H, \tag{1}$$

and then draw one $\omega$ randomly before the following steps. This defines one direction we want to optimize in the state space for the current run. The following three steps will then be performed in each iteration. We set the number of total iterations to be $T$.

#### 2.1.2 STEP 1: GUIDED TRANSITION SCORING

We first randomly select one position $i$ on the sequence so that we will update the token on this position during the current iteration. At each intermediate state $x_t$ and selected position $i$, each possible candidate transition $y^i \neq x^i$ is scored by combining local improvement measures with global directional alignment. The normalized rank score captures how much each individual objective improves relative to other possible token replacements, thereby encouraging exploration of promising local moves; formally, for each objective $n$ we compute

$$I_n(y^i, x) = \frac{\text{rank}\big(s_n(x_{\text{new}}) - s_n(x)\big)}{|\mathcal{T}|}, \tag{2}$$

where $x_{\text{new}}$ denotes the sequence obtained by replacing the $i$th token of $x$ with $y^i$. The $\text{rank}(\cdot)$ function maps the raw score change into a uniform scale in $[0, 1]$. In contrast, the directional term

$$D(y^i, x, \omega) = \Delta\mathbf{s}(y^i, x) \cdot \omega \tag{3}$$

measures the alignment of the multi-objective improvement vector $\Delta\mathbf{s} = [s_1(x_{\text{new}}) - s_1(x), s_2(x_{\text{new}}) - s_2(x), \cdots, s_n(x_{\text{new}}) - s_n(x)]$ with the chosen weight vector $\omega$, ensuring that transitions not only improve individual objectives but collectively move toward the desired trade-off direction. By z-score normalizing both components and combining them as

$$\Delta S(y^i, x, \omega) = \text{Norm}\Big[ \frac{1}{N} \sum_{n=1}^{N} I_n(y^i, x) \Big] + \lambda \, \text{Norm}\big[D(y^i, x, \omega)\big], \tag{4}$$

we balance rank-based exploration against direction-guided exploitation with $\lambda > 0$. Finally, we re-weight the original factorized velocity field from the pre-trained discrete flow matching model:

$$u^i_{\text{guided},t}(y^i, x \mid \omega) = \begin{cases} \beta\, u^i_t(y^i, x)\, \exp\big(\Delta S(y^i, x, \omega)\big), & y^i \neq x^i \\ -\sum_{y^i \neq x^i} u^i_{\text{guided},t}(y^i, x \mid \omega), & y^i = x^i \end{cases} \tag{5}$$

where $\beta$ is the strength hyperparameter. Therefore, the guided velocities satisfy the non-negativity and zero-sum rate conditions by construction, preserving valid CTMC dynamics while favoring high-utility transitions.

### 2.1.3 STEP 2: ADAPTIVE HYPERCONE FILTERING

To ensure each candidate token replacement drives the sequence towards the chosen trade-off direction, we restrict candidate transitions to lie within a cone around the weight vector $\omega$. This "hypercone" mechanism allows the sampler to navigate non-convex or discontinuous regions of the Pareto front by enforcing local directional consistency (Yuan et al., 2024). Specifically, for a given position $i$ and candidate token $y^i$, we compute the angle

$$\alpha^i = \arccos\left(\frac{\Delta\mathbf{s}(y^i, x) \cdot \omega}{\|\Delta\mathbf{s}(y^i, x)\|\, \|\omega\|}\right), \tag{6}$$

where $\Delta\mathbf{s}(y^i, x)$ is the multi-objective improvement vector from replacing $x^i$ with $y^i$. We accept only those $y^i$ for which $\alpha^i \leq \Phi$, where $\Phi$ denotes the current hypercone angle. Denoting $Y^i \subseteq T \setminus \{x^i\}$ as the set of accepted tokens, we select the best transition as

$$y^i_{\text{best}} = \arg \max_{y^i \in Y^i} \Delta S(y^i, x, \omega) \quad \text{if } Y^i \neq \emptyset. \tag{7}$$

There are two degenerate cases that may lead to empty $Y^i$: If every $\alpha^i \geq \pi$, indicating that all possible transitions decrease performance, we will perform a self-transition and retain the current state; if there exist some $\alpha^i < \pi$ but none lie within the cone (i.e. $\Phi$ is temporarily too small), we still advance by choosing the best-aligned candidate

$$y^i_{\text{best}} = \arg \max_{\{y' : \alpha^i < \pi\}} \Delta S(y^i, x, \omega), \tag{8}$$

allowing progress while the hypercone angle self-adjusts.

As a pre-defined hypercone angle may be too big or too small during the dynamic optimization process, we need to adaptively tune the angle for best balancing exploration and exploitation. Specifically, we compute the rejection rate

$$r_t = \frac{\#\{y^i : \alpha^i > \Phi\}}{\text{total \# of candidate transitions}} \tag{9}$$

and its exponential moving average (EMA)

$$\bar{r}_t = \alpha_r\, \bar{r}_{t-h} + \big(1 - \alpha_r\big)\, r_t, \tag{10}$$

where $\alpha_r \in [0, 1)$ is a smoothing coefficient and $\bar{r}_0 = \tau$ is the target rejection rate. We then update the hypercone angle via

$$\Phi_{t+h} = \text{clip}\Big(\Phi_t\, \exp\big(\eta\, (\bar{r}_t - \tau)\big),\, \Phi_{\min},\, \Phi_{\max}\Big), \tag{11}$$

with learning rate $\eta > 0$ and bounds $\Phi_{\min}, \Phi_{\max}$ to prevent the hypercone from collapsing or over-expanding. Intuitively, if too many candidates are being rejected ($\bar{r}_t > \tau$), the hypercone widens to admit more directions; if too few are rejected ($\bar{r}_t < \tau$), it narrows to focus on the most aligned transitions.

### 2.1.4 STEP 3: EULER SAMPLING

Once the guided transition rates $u^i_{\text{guided},t}(y^i, x \mid \omega)$ have been computed and the best candidate transition has been selected after hypercone filtering (if not self-transitioning), we evolve the CTMC via Euler sampling. Specifically, we denote the total outgoing rate from $x$ at time $t$ on coordinate $i$ by

$$R^i_t(x) = -u^i_{\text{guided},t}(x^i, x \mid \omega) = \sum_{y^i \neq x^i} u^i_{\text{guided},t}(y^i, x \mid \omega). \tag{12}$$

Table 1: **Evaluation of unconditional EnhancerDNA generation.** Each method generates 10k sequences, and we compare their empirical distributions with the data distributions using the Fréchet Biological distance (FBD) metric. NFE refers to number of function evaluations. # Training Epochs refers to the number of training epochs needed to get the model checkpoint for this evaluation. The Random Sequence baseline shows the FBD for the same number and length of sequences with uniform randomly chosen nucleotides. Dirichlet FM refers to the Dirichlet Flow Matching model.

|  | **FBD** | **NFE** | **# Training Epochs** |
|---|---|---|---|
| Random Sequence | 622.8 | - | - |
| Dirichlet FM | 5.3 | 100 | 1400 |
| EnhancerDFM | 5.9 | 100 | 20 |

The one-step transition kernel for coordinate $i$ is given by the exact Euler-Maruyama analogue for CTMCs:

$$\mathbb{P}\big(X^i_{t+h} = y^i \mid X_t = x\big) = \begin{cases} \exp\big(h\, u^i_{\text{guided},t}(x^i, x \mid \omega)\big) = \exp\big(-h\, R^i_t(x)\big), & y^i = x^i, \\ \dfrac{u^i_{\text{guided},t}(y^i, x \mid \omega)}{R^i_t(x)}\big(1 - \exp(-h\, R^i_t(x))\big), & y^i \neq x^i. \end{cases} \tag{13}$$

Here, $h = 1/T$ is the step size in the time interval, $X_t$ and $X_{t+h}$ denotes the current state and the next state respectively. In practice, one draws a uniform random number $r \in [0,1]$: if $r \leq 1 - \exp\big(-h\, R^i_t(x)\big)$, $x^i$ will transition to the best selected candidate; otherwise we retain $x^i$.

After performing from step 1 to step 3 for $T$ iterations, we end with the final sample $x_1$ whose score vectors have been steered near the Pareto Front, with all objectives optimized.

## 3 EXPERIMENTS

To the best of our knowledge, there are no public datasets that serve to benchmark multi-objective optimization algorithms for biological sequences. Therefore, we benchmark MOG-DFM on two tasks: multi-objective guided peptide binder sequence generation and multi-objective guided enhancer DNA sequence generation. We will first show two discrete flow matching models developed for peptide generation and enhancer DNA generation, then we will demonstrate MOG-DFM's efficacy on a wide variety of tasks and examples.

### 3.1 PEPDFM AND ENHANCERDFM GENERATE DIVERSE AND BIOLOGICALLY PLAUSIBLE SEQUENCES

To enable the efficient generation of peptide binders, we developed **PepDFM**, an unconditional peptide generator based on the Discrete Flow Matching (DFM) framework (Gat et al., 2024) with a U-Net-style convolutional architecture (Ronneberger et al., 2015). Trained on a combined dataset from PepNN, BioLip2, and PPIRef Abdin et al. (2022); Zhang et al. (2024); Bushuiev et al. (2023), PepDFM achieved a validation loss of 3.1051. As described in Section A, the low generalized KL loss during evaluation demonstrates PepDFM's strong performance. PepDFM can generate diverse and novel peptides, shown by high Hamming distances from the test set, while the Shannon entropy of PepDFM-generated samples matches the test set, confirming the biological plausibility of the generated sequences (Figure A1).

EnhancerDFM adopts the same model backbone and melanoma enhancer dataset used in Enhancer DNA design task from Stark, et al. Stark et al. (2024). We employed the Fréchet Biological distance (FBD) metric from Stark et al. (2024) to evaluate the performance of EnhancerDFM (Table 1). Specifically, using the same number of function evaluations (NFE), EnhancerDFM achieved a comparable FBD of 5.9 compared with Dirichlet FM of 5.3, significantly lower than the FBD of random sequences, demonstrating EnhancerDFM's ability to design biologically plausible enhancer DNA sequences. Significantly, the best EnhancerDFM model is achieved within 20 training epochs, while the best Dirichlet FM is obtained only in around 1400 training epochs, highlighting discrete flow matching models' superior capability of capturing the underlying data distribution.

Table 2: MOG-DFM generates peptide binders for 10 diverse protein targets, optimizing five therapeutic properties: hemolysis, non-fouling, solubility, half-life (in hours), and binding affinity. Each value represents the average of 100 MOG-DFM-designed binders.

| Target | Binder Length | Hemolysis (↓) | Non-Fouling (↑) | Solubility (↑) | Half-Life (↑) | Affinity (↑) |
|--------|--------|--------|--------|--------|--------|--------|
| AMHR2 | 8 | 0.0755 | 0.8352 | 0.8219 | 31.624 | 7.3789 |
| AMHR2 | 12 | 0.0570 | 0.8419 | 0.8279 | 28.761 | 7.4274 |
| AMHR2 | 16 | 0.0618 | 0.7782 | 0.7428 | 31.227 | 7.6099 |
| EWS::FLI1 | 8 | 0.0809 | 0.8508 | 0.8296 | 47.169 | 6.2251 |
| EWS::FLI1 | 12 | 0.0616 | 0.8302 | 0.8130 | 34.225 | 6.3631 |
| EWS::FLI1 | 16 | 0.0709 | 0.7787 | 0.7400 | 34.192 | 6.5912 |
| MYC | 8 | 0.0809 | 0.8135 | 0.8005 | 39.836 | 6.8488 |
| OX1R | 10 | 0.0741 | 0.8115 | 0.7969 | 33.533 | 7.4162 |
| DUSP12 | 9 | 0.0735 | 0.8360 | 0.8216 | 33.754 | 6.4946 |
| 1B8Q | 8 | 0.0744 | 0.8334 | 0.827 | 33.243 | 5.932 |
| 1E6I | 6 | 0.0887 | 0.7884 | 0.7793 | 41.164 | 4.9621 |
| 3IDJ | 7 | 0.0924 | 0.8246 | 0.7992 | 30.388 | 7.6304 |
| 5AZ8 | 11 | 0.0698 | 0.8462 | 0.8420 | 28.726 | 6.6051 |
| 7JVS | 11 | 0.0628 | 0.8390 | 0.8206 | 32.834 | 6.9569 |

## 3.2 MOG-DFM EFFECTIVELY BALANCES EACH OBJECTIVE TRADE-OFF

To validate that MOG-DFM framework can balance the trade-offs between each objective, we performed two sets of experiments for peptide binder generation with three property guidance, and in ablation experiment settings, we removed one or more objectives. In the binder design task for target 7LUL (affinity, solubility, hemolysis guidance; Table 7), omitting any single guidance causes a collapse in that property, while the remaining guided metrics may modestly improve. Likewise, in the binder design task for target CLK1 (affinity, non-fouling, half-life guidance; Table 8), disabling non-fouling guidance allows half-life to exceed 80 hours but drives non-fouling near zero, and disabling half-life guidance preserves non-fouling yet reduces half-life below 2 hours. In contrast, enabling all guidance signals produces the most balanced profiles across all objectives. These results confirm that MOG-DFM precisely targets chosen objectives while preserving the flexibility to navigate conflicting requirements and push samples toward the Pareto front, thereby demonstrating the correctness and precision of our multi-objective sampling framework.

## 3.3 MOG-DFM GENERATES PEPTIDE BINDERS UNDER FIVE PROPERTY GUIDANCE

We next benchmark MOG-DFM on a peptide binder generation task guided by five different properties that are critical for therapeutic discovery: hemolysis, non-fouling, solubility, half-life, and binding affinity. To evaluate MOG-DFM in a controlled setting, we designed 100 peptide binders per target for ten diverse proteins, structured targets with known binders (1B8Q, 1E6I, 3IDJ, 5AZ8, 7JVS), structured targets without known binders (AMHR2, OX1R, DUSP12), and intrinsically disordered targets (EWS::FLI1, MYC) (Table 2). Across all targets and across multiple binder lengths, the generated peptides achieve low hemolysis rates (0.06-0.09), high non-fouling (>0.78) and solubility (>0.74), extended half-life (28-47 h), and strong affinity scores (6.4-7.6), demonstrating both balanced optimization and robustness to sequence length.

For the target proteins with pre-existing binders, we compared the property values between their known binders with MOG-DFM-designed ones (Figure 2A,B, A2). The designed binders significantly outperform the pre-existing binders across all properties without compromising the binding potential, which is further confirmed by the ipTM scores computed by AlphaFold3 (Abramson et al., 2024) and docking scores calculated by AutoDock VINA (Trott & Olson, 2010). Although the MOG-DFM-designed binders bind to similar target positions as the pre-existing ones, they differ significantly in sequence and structure, demonstrating MOG-DFM's capacity to explore the vast sequence space for optimal designs. For target proteins without known binders, complex structures were visualized using one of the MOG-DFM-designed binders (Figure A3). The corresponding property scores, as well as ipTM and docking scores, are also displayed. Some of the designed binders showed longer half-life,

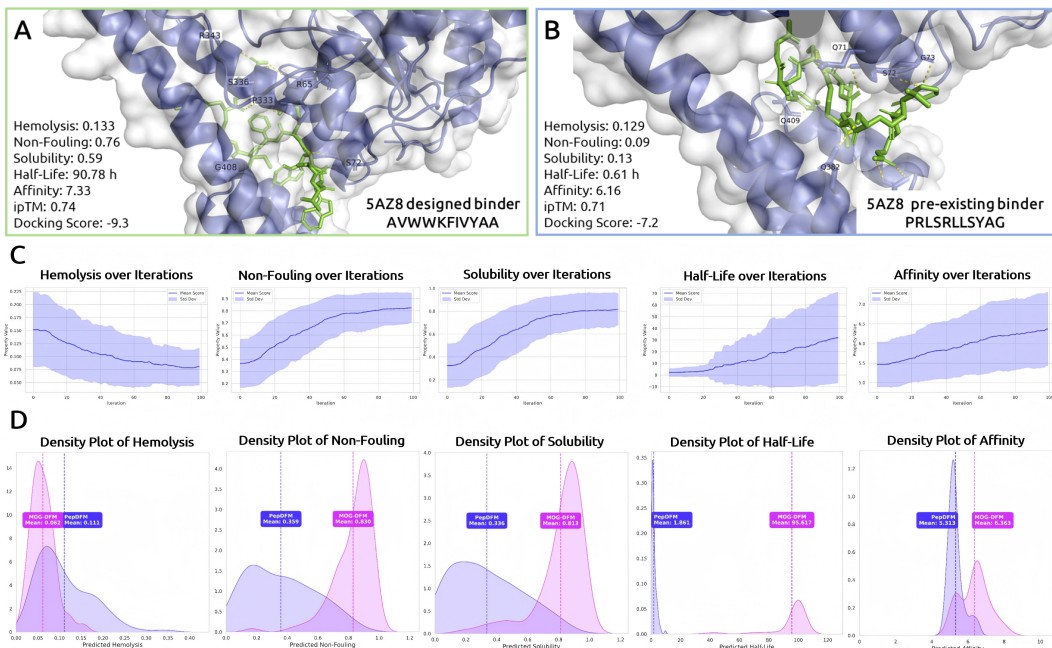

Figure 2: **(A), (B)** Complex structures of PDB 5AZ8 with a MOG-DFM-designed binder and its pre-existing binder. Five property scores are shown for each binder, along with the ipTM score from AlphaFold3 and docking score from AutoDock VINA. Interacting residues on the target are visualized. **(C)** Plots showing the mean scores for each property across the number of iterations during MOG-DFM's design of binders of length 12-aa for EWS::FLI1. **(D)** Density plots illustrating the distribution of predicted property scores for MOG-DFM-designed EWS::FLI1 binders of length 12-aa, compared to the peptides generated unconditionally by PepDFM. Please zoom in for better viewing.

while others excelled in non-fouling and solubility, underscoring the comprehensive exploration of the sequence space by MOG-DFM.

At each iteration, we recorded the mean and standard deviation of the five property scores across all the 100 binders to evaluate the effectiveness of the guided generation strategy (Figure 2C). All five properties exhibited an improving trend over iterations, with the average score of the solubility and non-fouling properties showing a significant increase from a score around 0.3 to 0.8. A large deviation of the final half-life values is caused by the susceptibility of the half-life value to guidance, with MOG-DFM balancing the trade-offs between half-life and other values. The improvements of hemolysis, non-fouling, and solubility gradually converge, demonstrating MOG-DFM's efficiency in steering the generation process to the Pareto Front within only 100 iterations.

We visualized the distribution change steered by MOG-DFM by plotting the property score distribution of 100 peptides of length 12 designed for EWS::FLI1 and 100 peptides of the same length sampled unconditionally from PepDFM (Figure 2D). MOG-DFM effectively shifted and concentrated the peptide distribution so that the peptides possess improved properties for all the objectives, demonstrating MOG-DFM's ability to steer the generation so that all properties are optimized simultaneously.

In Section C, we demonstrate the reliability of our score models. We now use external evaluation tools to further confirm that MOG-DFM-designed binders possess desired properties. The average solubility and half-life for each target across all 100 designed peptides were predicted using ADMET-AI (Table 5) (Swanson et al., 2024). ADMET-AI, trained on a different dataset from our solubility and half-life prediction models, predicts average LogS values around -2.5 log mol·L$^{-1}$, which is well above the conventional -4 threshold for good solubility, and confirms long half-life estimates (>15 h). These results from an orthogonal predictive model demonstrate MOG-DFM's capability to generate candidates with multiple desirable drug properties.

Table 3: MOG-DFM outperforms traditional multi-objective optimization algorithms in designing peptide binders guided by five objectives. Each value represents the average of 100 designed binders. The table also records the average runtime for each algorithm to design a single binder. The best result for each metric is highlighted in bold.

| Target | Method | Time (s) | Hemolysis (↓) | Non-Fouling | Solubility | Half-Life | Affinity |
|--------|--------|----------|---------------|-------------|------------|-----------|----------|
| 1B8Q | MOPSO | 8.54 | 0.1066 | 0.4763 | 0.4684 | 4.449 | 6.0594 |
| | NSGA-III | 33.13 | 0.0862 | 0.5715 | 0.5825 | 7.324 | 7.2178 |
| | SMS-EMOA | 8.21 | 0.1196 | 0.3450 | 0.3511 | 3.023 | 5.955 |
| | SPEA2 | 17.48 | 0.0819 | 0.4973 | 0.5057 | 4.126 | **7.324** |
| | PepTune + DPLM | **2.46** | 0.1453 | 0.3085 | 0.3213 | 1.1737 | 5.2398 |
| | **MOG-DFM** | 43.00 | **0.0785** | **0.8445** | **0.8455** | **27.227** | 5.9094 |
| PPP5 | MOPSO | 11.34 | 0.0883 | 0.4711 | 0.4255 | 1.769 | 6.6958 |
| | NSGA-III | 37.30 | **0.0479** | 0.7138 | 0.7066 | 2.901 | 7.3789 |
| | SMS-EMOA | 8.43 | 0.1242 | 0.4269 | 0.4334 | 1.031 | 6.2854 |
| | SPEA2 | 19.02 | 0.0555 | 0.6221 | 0.6098 | 2.613 | **7.6253** |
| | PepTune + DPLM | **4.80** | 0.1184 | 0.2752 | 0.2636 | 1.2667 | 5.8454 |
| | **MOG-DFM** | 90.00 | 0.0617 | **0.7738** | **0.751** | **27.775** | 6.8197 |

## 3.4 MOG-DFM OUTPERFORMS STATE-OF-THE-ART BASELINES

We benchmarked MOG-DFM against four established multi-objective optimization (MOO) baselines (NSGA-III (Deb & Jain, 2013), SMS-EMOA (Beume et al., 2007), SPEA2 (Zitzler et al., 2001), and MOPSO (Coello & Lechuga, 2002)) on two protein targets: 1B8Q, a small protein with known peptide binders, and PPP5, a larger protein without characterized binders (Table 3). Each method generated 100 candidate binders optimized for five properties: hemolysis, non-fouling, solubility, half-life, and binding affinity. While MOG-DFM required longer runtimes than evolutionary baselines, it consistently produced the best trade-offs. For both targets, it lowered hemolysis by more than 10%, increased non-fouling and solubility by 30-50%, and extended half-life by a factor of 3 to 4 relative to the next-best method, all while maintaining competitive binding affinity. These results underscore MOG-DFM's effectiveness in navigating high-dimensional property landscapes to yield peptide binders with balanced, optimized profiles.

We also compared against PepTune (Tang et al., 2025b), a recent masked discrete diffusion model for peptide design that couples generation with Monte Carlo Tree Search for MOO. PepTune's backbone was adapted to the existing DPLM model Wang et al. (2024) for wild-type peptide sequence generation. Despite longer runtimes, MOG-DFM substantially outperformed PepTune across all objectives, yielding nearly threefold improvements in non-fouling and solubility and a 22-fold increase in half-life. Together, these comparisons demonstrate that MOG-DFM surpasses not only traditional MOO algorithms but also the current state-of-the-art diffusion-based approach for multi-objective-guided peptide binder design.

## 3.5 MOG-DFM GENERATES ENHANCER DNA OF SPECIFIC CLASS WITH SPECIFIED DNA SHAPES

To demonstrate the universal capability of MOG-DFM in performing multi-objective guided generation for biological sequences, we applied MOG-DFM to design enhancer DNA sequences guided by enhancer class and DNA shape. EnhancerDFM was used as the unconditional enhancer DNA sequence generator, while Deep DNAshape was employed to predict DNA shape (Li et al., 2024), and the enhancer class predictor was sourced from Stark et al. (2024). Two distinct tasks with different enhancer class and DNA shape guidance were carried out, and ablation results are presented in Table 9. Given the time constraints, we designed five enhancer sequences of length 100 for each setting.

In the first task, we conditioned the generation to target enhancer class 1 (associated with the transcription factor binding motif ATF) and a high HelT (helix twist) value, with the maximum HelT value set to 36. With both guidance criteria in place, MOG-DFM effectively steered the sequence generation towards enhancer class 1 while simultaneously ensuring that the HelT value approached its maximum (Table 9). When one or both guidance criteria were removed, the corresponding properties

showed significant degradation, with the probability of achieving the desired enhancer class dropping near zero (Table 9). A similar outcome was observed in the second task, which targeted enhancer class 16 and a higher Rise shape value, with the maximum Rise value set to 3.7. Since the canonical range for the Rise shape value spans from 3.3 to 3.4, MOG-DFM ensured both a high probability for the target enhancer class and an optimal DNA shape value, outperforming other ablation settings (Table 9).

## 4 RELATED WORKS

**Online Multi-Objective Optimization.** Recent advances in multi-objective guided generation have focused on online or sequential decision-making, where solutions are refined with newly acquired data (Gruver et al., 2023; Jain et al., 2023; Stanton et al., 2022; Ahmadianshalchi et al., 2024). A prominent approach is Bayesian optimization (BO), which iteratively builds a probabilistic surrogate from observed data and uses an acquisition function to propose the next evaluation (Yu et al., 2020; Shahriari et al., 2015). Multi-objective BO often reduces the problem via scalarization (Knowles, 2006; Zhang & Li, 2007; Paria et al., 2020), or employs more advanced acquisition criteria such as expected hypervolume improvement (EHVI) (Emmerich & Klinkenberg, 2008) or information gain (Belakaria et al., 2021). While MOG-DFM's directional guidance resembles scalarization and its Pareto coverage aligns with hypervolume principles, our framework addresses therapeutic design in an offline regime where each sequence requires costly wet-lab or high-fidelity in-silico evaluation. This offline, single-batch generation contrasts with the sequential, feedback-driven nature of online methods, making direct numerical comparison inappropriate.

**Offline Multi-Objective Optimization with Diffusion and Flow Matching.** Recent advances in generative modeling, particularly with diffusion and flow matching, have introduced powerful new tools for multi-objective optimization. For instance, ParetoFlow leverages a flow matching model guided by a weighted scalarization of the objectives, while PGD-MOO uses a preference-based classifier to guide a diffusion model towards the Pareto front (Yuan et al., 2024; Annadani et al., 2025). However, a key characteristic of these methods is that they are designed to operate in continuous design or latent spaces. In contrast, MOG-DFM is specifically developed for the discrete token space inherent to biological sequences. This fundamental difference in the underlying data domain, continuous vectors versus discrete sequences, precludes a direct and meaningful benchmark.

**Offline Multi-Objective Frameworks for Biomolecule Generation.** Recent efforts in offline multi-objective optimization have also targeted biomolecule generation. Methods like EGD and MUDM have shown success in designing molecules with multiple optimized properties (Sun et al., 2025; Han et al., 2023). However, these approaches primarily focus on generating or optimizing based on 3D structural representations of proteins. In contrast, our MOG-DFM framework is a sequence-only algorithm that operates directly in the discrete space of amino acids or nucleotides. This fundamental difference in data modality makes them unsuitable for a direct numerical benchmark.

## 5 DISCUSSION

In this work, we have presented **Multi-Objective-Guided Discrete Flow Matching (MOG-DFM)**, a scalable framework for generating biomolecular sequences that simultaneously optimize multiple, often conflicting properties. By guiding discrete flow matching models with multi-objective optimization, MOG-DFM enables the design of peptide and DNA sequences with improved therapeutic and structural characteristics. While MOG-DFM performs well in biological domains, it may face challenges when scaling to longer sequences, both due to increased computational complexity and potentially slower convergence. Future work will therefore focus on extending the framework to handle longer peptides and higher-dimensional outputs, including applications in text and image generation. From a theoretical perspective, improving Pareto-convergence guarantees and incorporating uncertainty-aware or feedback-driven guidance remain key directions. Ultimately, MOG-DFM offers a foundation for generating the next generation of therapeutics, molecules that are not only effective but explicitly optimized for the multifaceted properties critical to clinical success.

## REPRODUCIBILITY STATEMENT

We have ensured reproducibility through detailed theoretical, algorithmic, and experimental descriptions. The full MOG-DFM procedure is formalized in Section A and illustrated schematically in Figure 1. Proofs of theoretical guarantees are provided in Section G. Base discrete flow matching models (PepDFM and EnhancerDFM) are described with complete architecture, training details, and datasets in Appendix Sections A and C, with quantitative metrics reported in Table 1. Hyperparameter sensitivity benchmarks are provided in Table 10, and additional ablation studies are presented throughout the Appendix. Benchmark comparisons against classical and diffusion-based baselines are reported in Tables 3 and 5. All datasets (PepNN, BioLip2, PPIRef, melanoma enhancer sequences, PepLand, PeptideBERT, PEPLife, PepTherDia, THPdb2) are publicly available. We will release code, trained PepDFM/EnhancerDFM checkpoints, and MOG-DFM sampling scripts to enable full reproducibility.

## ETHICS STATEMENT

This work develops a general generative modeling framework for multi-objective sequence design, with applications demonstrated on peptides and enhancer DNA. All data used are publicly available and non-sensitive, consisting of peptide property datasets, enhancer DNA sequences, and benchmark protein–peptide interaction sets. No human subjects, patient data, or animal experiments were involved. Potential risks include the misuse of generative models for harmful molecule design. To mitigate these risks, we will release code and pretrained models strictly under a research-only license, and provide documentation emphasizing safe and responsible use. The societal benefits (improving therapeutic peptide design, enhancing drug safety profiles, and enabling efficient exploration of biological sequence space) substantially outweigh potential risks. We encourage future work using MOG-DFM to adhere to similar safeguards.

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

## A    DISCRETE FLOW MATCHING

In this paper, the notation for discrete flow matching follows Gat et al. (2024). In the discrete setting, we consider data $x = (x_1, \ldots, x_d)$ taking values in a finite state space $S = \mathcal{T}^d$, where $\mathcal{T} = [K] = \{1, 2, \ldots, K\}$ is called the vocabulary. We model a continuous-time Markov chain (CTMC) $\{X_t\}_{t \in [0,1]}$ whose time-dependent transition rates $u_t(y, x)$ transport probability mass from an initial distribution $p_0$ to a target distribution $p_1$ (Gat et al., 2024). The marginal probability at time $t$ is denoted $p_t(x)$, and its evolution is governed by the Kolmogorov forward equation

$$\frac{\mathrm{d}}{\mathrm{d}t} p_t(y) = \sum_{x \in S} u_t(y, x) \, p_t(x), \tag{14}$$

where $y$ is transitioned sequence. The learnable velocity field $u_t(y, x)$ is defined as the sum of factorized velocities:

$$u_t(y, x) = \sum_i \delta(y^{\bar{i}}, x^{\bar{i}}) u_t^i(y^i, x), \tag{15}$$

where $\bar{i} = (1, \ldots, i-1, i+1, \ldots, d)$ denotes all indices excluding $i$ and $\delta$ is the Kronecker delta function. The rate conditions for factorized velocities $u_t^i(y^i, x)$ are required per dimension $i \in [d]$:

$$u_t(y, x) \geq 0 \text{ for all } y^i \neq x^i, \text{ and } \sum_{y^i \in \mathcal{T}} u_t^i(y^i, x) = 0 \text{ for all } x \in S, \tag{16}$$

so that for small $h > 0$, the one-step kernel

$$p_{t+h|t}(y \mid x) = \delta(y, x) + h \, u_t(y, x) + o(h) \tag{17}$$

remains a proper probability mass function.

The goal of training a discrete flow matching model is to learn the velocity field $u_t^\theta$. Representing the marginal velocity $u_t^\theta$ in terms of factorized velocities $u_t^{\theta,i}$ enables the following conditional flow matching loss

$$\mathcal{L}_{\mathrm{CDFM}}(\theta) = \mathbb{E}_{t, Z, X_t \sim p_{t|Z}} \sum_i D_{X_t}^i \left( u_t^i(\cdot, X_t \mid Z), u_t^{\theta,i}(\cdot, X_t) \right), \tag{18}$$

where $t \sim \mathcal{U}[0,1]$, $Z$ represents a random variable, and $u_t^i(\cdot, x \mid z), u_t^{\theta,i}(\cdot, x) \in \mathbb{R}^\mathcal{T}$ satisfy the rate conditions. This means that $u_t^i(\cdot, x \mid z), u_t^{\theta,i}(\cdot, x) \in \Omega_{x^i}$ where, for $\alpha \in \mathcal{T}$, we define

$$\Omega_\alpha = \left\{ v \in \mathbb{R}^\mathcal{T} \;\middle|\; v(\beta) \geq 0 \; \forall \beta \in \mathcal{T} \setminus \{\alpha\}, \text{ and } v(\alpha) = -\sum_{\beta \neq \alpha} v(\beta) \right\} \subset \mathbb{R}^\mathcal{T}. \tag{19}$$

This is a convex set, and $D_x^i(u, v)$ is a Bregman divergence defined by a convex function $\Phi_x^i : \Omega_{x^i} \to \mathbb{R}$.

In practice, we can further parametrize the velocity field using a mixture path. Specifically, one defines a mixture path with scheduler $\kappa_t \in [0, 1]$ so that each coordinate $X_t^{(i)}$ equals $x_0^{(i)}$ or $x_1^{(i)}$ with probabilities $1 - \kappa_t$ and $\kappa_t$ respectively. The mixture marginal velocity is then obtained by averaging the conditional rates over the posterior of $(x_0, x_1)$ given $X_t = x$, yielding

$$u_t^i(y^i, x) = \sum_{x_1^i} \frac{\dot{\kappa}_t}{1 - \kappa_t} \left[ \delta(y^i, x_1^i) - \delta(y^i, x^i) \right] p_{1|t}^i(x_1^i \mid x), \tag{20}$$

where $\dot{\kappa}_t$ denotes the time derivative of $\kappa_t$. Therefore, the aim of discrete flow matching model training, which is to learn the velocity field $u_t^i(y^i, x)$, now equals to learning the marginal posterior $p_{1|t}^i(x_1^i \mid x)$. In this case, we can set the Bregman divergence to the generalized KL comparing general vectors $u, v \in \mathbb{R}_{\geq 0}^m$,

$$D(u, v) = \sum_j \left[ u_j \log \frac{u_j}{v_j} - u_j + v_j \right]. \tag{21}$$

For this choice of $D$, we get

$$D\left(u_t^i(\cdot, x^i \mid x_0, x_1), u_t^{\theta,i}(\cdot, x)\right) = \frac{\dot{\kappa}_t}{1 - \kappa_t}\left[(\delta(x_1^i, x^i) - 1)\log p_{1|t}^{\theta,i}(x_1^i \mid x) + \delta(x_1^i, x^i) - p_{1|t}^{\theta,i}(x^i \mid x)\right] \tag{22}$$

which implements the loss (8) when conditioning on $Z = (X_0, X_1)$. The generalized KL loss also provides an evidence lower bound (ELBO) on the likelihood of the target distribution

$$-\log p_1^\theta(x_1) \leq \mathbb{E}_{t, X_0, X_t \sim p_{t|0,1}} \sum_i D\left(u_t^i(\cdot, X_1^i \mid X_0, x_1), u_t^{\theta,i}(\cdot, X_t)\right), \tag{23}$$

where $p_1^\theta$ is the marginal generated by the model at time $t = 1$. Therefore, in addition to training, the generalized KL loss can also be used for evaluation.

## B  BASE MODEL DETAILS

### B.1  PEPDFM

**Model Architecture.** The base model is a time-dependent architecture based on U-Net (Ronneberger et al., 2015). It uses two separate embedding layers for sequence and time, followed by five convolutional blocks with varying dilation rates to capture temporal dependencies, while incorporating time-conditioning through dense layers. The final output layer generates logits for each token. We used a polynomial convex schedule with a polynomial exponent of 2.0 for the mixture discrete probability path in the discrete flow matching.

**Dataset Curation.** The dataset for PepDFM training was curated from the PepNN, BioLip2, and PPIRef dataset (Abdin et al., 2022; Zhang et al., 2024; Bushuiev et al., 2023). All peptides from PepNN and BioLip2 were included, along with sequences from PPIRef ranging from 6 to 49 amino acids in length. The dataset was divided into training, validation, and test sets at an 80/10/10 ratio.

**Training Strategy.** The training is conducted on a 2xH100 NVIDIA NVL GPU system with 94 GB of VRAM for 200 epochs with batch size 512. The model checkpoint with the lowest evaluation loss was saved. The Adam optimizer was employed with a learning rate 1e-4. A learning rate scheduler with 20 warm-up epochs and cosine decay was used, with initial and minimum learning rates both 1e-5. The embedding dimension and hidden dimension were set to be 512 and 256 respectively for the base model.

**Dynamic Batching.** To enhance computational efficiency and manage variable-length token sequences, we implemented dynamic batching. Drawing inspiration from ESM-2's approach (Lin et al., 2023), input peptide sequences were sorted by length to optimize GPU memory utilization, with a maximum token size of 100 per GPU.

### B.2  ENHANCERDFM

**Model Architecture.** The base model for EnhancerDFM applies the same architecture as the PepDFM. We also used a polynomial convex schedule with a polynomial exponent of 2.0 for the mixture discrete probability path in the discrete flow matching.

**Dataset Curation.** The dataset for EnhancerDFM training is curated by (Stark et al., 2024). The dataset contains 89k enhancer sequences from human melanoma cells (Atak et al., 2021). Each sequence is of length 500 paired with cell class labels determined from ATAC-seq data (Buenrostro et al., 2013). There are 47 such classes of cells in total, with details displayed in Table 11 (Atak et al., 2021). We applied the same dataset split strategy as (Stark et al., 2024).

**Training Strategy.** The training is conducted on a 2xH100 NVIDIA NVL GPU system with 94 GB of VRAM for 1500 epochs with batch size 256. The model checkpoint with the lowest evaluation loss was saved. The Adam optimizer was employed with a learning rate 1e-3. A learning rate scheduler with 150 warm-up epochs and cosine decay was used, with initial and minimum learning rates both 1e-4. Both the embedding dimension and hidden dimension were set to be 256 for the base model.

# C   SCORE MODEL DETAILS

We collected hemolysis (9,316), non-fouling (17,185), solubility (18,453), and binding affinity (1,781) data for classifier training from the PepLand and PeptideBERT datasets (Zhang et al., 2023; Guntuboina et al., 2023). All sequences taken are wild-type L-amino acids and are tokenized and represented by ESM-2 protein language model Lin et al. (2023).

## C.1   BOOSTED TREES FOR CLASSIFICATION

For hemolysis, non-fouling, and solubility classification, we trained XGBoost boosted tree models for logistic regression. We split the data into 0.8/0.2 train/validation using stratified splits from scikit-learn Pedregosa et al. (2011) and generated mean pooled ESM-2-650M Lin et al. (2023) embeddings as input features to the model. We ran 50 trials of OPTUNA Akiba et al. (2019) search to determine the optimal XGBoost hyperparameters (Table 4) tracking the best binary classification F1 scores. The best models for each property reached F1 scores of: 0.58, 0.71, and 0.68 on the validation sets accordingly.

Table 4: XGBoost Hyperparameters for Classification

| Hyperparameter | Value/Range |
|---|---|
| Objective | `binary:logistic` |
| Lambda | $[1e{-}8, 10.0]$ |
| Alpha | $[1e{-}8, 10.0]$ |
| Colsample by Tree | $[0.1, 1.0]$ |
| Subsample | $[0.1, 1.0]$ |
| Learning Rate | $[0.01, 0.3]$ |
| Max Depth | $[2, 30]$ |
| Min Child Weight | $[1, 20]$ |
| Tree Method | `hist` |

## C.2   BINDING AFFINITY SCORE MODEL

We developed an unpooled reciprocal attention transformer model to predict protein-peptide binding affinity, leveraging latent representations from the ESM-2 650M protein language model Lin et al. (2023). Instead of relying on pooled representations, the model retains unpooled token-level embeddings from ESM-2, which are passed through convolutional layers followed by cross-attention layers. The binding affinity data was split into a 0.8/0.2 ratio, maintaining similar affinity score distributions across splits. We used OPTUNA Akiba et al. (2019) for hyperparameter optimization tracing validation correlation scores. The final model was trained for 50 epochs with a learning rate of 3.84e-5, a dropout rate of 0.15, 3 initial CNN kernel layers (dimension 384), 4 cross-attention layers (dimension 2048), and a shared prediction head (dimension 1024) in the end. The classifier reached 0.64 Spearman's correlation score on validation data.

## C.3   HALF-LIFE SCORE MODEL

**Dataset Curation.** The half-life dataset is curated from three publicly available datasets: PEPLife, PepTherDia, and THPdb2 (Mathur et al., 2016; D'Aloisio et al., 2021; Jain et al., 2024). Data related to human subjects were selected, and entries with missing half-life values were excluded. After removing duplicates, the final dataset consists of 105 entries.

**Pre-training on stability data.** Given the small size of the half-life dataset, which is insufficient for training a model to capture the underlying data distribution, we first pre-trained a score model on a larger stability dataset to predict peptide stability (Tsuboyama et al., 2023). The model consists of three linear layers with ReLU activation functions, and a dropout rate of 0.3 was applied. The

model was trained on a 2xH100 NVIDIA NVL GPU system with 94 GB of VRAM for 50 epochs. The Adam optimizer was employed with a learning rate 1e-2. A learning rate scheduler with 5 warm-up epochs and cosine decay was used, with initial and minimum learning rates both 1e-3. After training, the model achieved a validation Spearman's correlation of 0.7915 and an $R^2$ value of 0.6864, demonstrating the reliability of the stability score model.

**Fine-tuning on half-life data.** The pre-trained stability score model was subsequently fine-tuned on the half-life dataset. Since half-life values span a wide range, the model was adapted to predict the base-10 logarithm of the half-life (h) values to stabilize the learning process. After fine-tuning, the model achieved a validation Spearman's correlation of 0.8581 and an $R^2$ value of 0.5977.

## D  SAMPLING DETAILS

### D.1  PEPTIDE BINDER GENERATION TASKS

**Objective Description.** Five key property objectives are considered in the peptide binder generation tasks: hemolysis, non-fouling, solubility, half-life, and binding affinity. Each of these properties plays a crucial role in optimizing the therapeutic potential of peptides. Hemolysis refers to the peptide's ability to minimize red blood cell lysis, ensuring safe systemic circulation (Pirtskhalava et al., 2013). Non-fouling properties describe the peptide's resistance to unwanted interactions with biomolecules, thus enhancing its stability and bioavailability in vivo (Chen et al., 2009). Solubility is critical for ensuring adequate peptide dissolution in biological fluids, directly influencing its absorption and therapeutic efficacy (Fosgerau & Hoffmann, 2015). Half-life indicates the duration for which the peptide remains active in circulation, which is vital for reducing dosing frequency (Swanson, 2014). Finally, binding affinity measures the strength of the peptide's interaction with its target, directly correlating to its biological activity and potency in therapeutic applications (Bostrom et al., 2008).

**Score Model Settings.** To align all objectives as maximization, we convert the predicted hemolysis rate $h$ into a score $1 - h$, so that lower hemolysis yields a higher value. We also cap the predicted log-scale half-life at 2 (i.e., 100 h) to prevent it from dominating the optimization and ensure balanced trade-offs across all properties. For the remaining objectives, non-fouling, solubility, and binding affinity, we directly employ their model outputs during sampling.

**Hyperparameter Settings.** The hyperparameters were set as follows: The number of divisions used in generating weight vectors, num_div, was set to 64, $\lambda$ to 1.0, $\beta$ to 1.0, $\alpha_r$ to 0.5, $\tau$ to 0.3, $\eta$ to 1.0, $\Phi_{init}$ to 45°, $\Phi_{min}$ to 15°, $\Phi_{max}$ to 75°. The total sampling step $T$ was 100.

### D.2  ENHANCER DNA GENERATION TASKS

**Hyperparameter Settings.** The hyperparameters were set the same as those in peptide binder generation tasks, except that the total sampling step $T$ was set to 800.

## E  HYPERPARAMETER SENSITIVITY BENCHMARK

There are several hyperparameters in MOG-DFM whose settings may affect generative performance. To assess this sensitivity, we evaluated peptide binder design across a broad range of values for each parameter (Table 10). We find that increasing the number of sampling steps consistently improves all performance metrics, as finer discretization more closely approximates the continuous-time dynamics. In contrast, setting the initial hypercone angle $\Phi_{init}$ too small or too large both degrade results: an overly narrow cone restricts exploration, while an overly wide cone dilutes directional guidance. By comparison, the remaining hyperparameters (i.e., $\beta$, $\lambda$, $\alpha_r$, $\eta$, $\tau$, and the bounds $\Phi_{min}$, $\Phi_{max}$) exhibit only modest impact on outcomes, indicating that MOG-DFM is robust to moderate variations in these settings.

## F  ADAPTIVE HYPERCONE FILTERING ENHANCES MULTI-OBJECTIVE OPTIMIZATION

To quantify the contribution of our adaptive hypercone mechanism, we performed an ablation study on three protein targets (3IDJ, 4E-BP2, and EWS::FLI1), generating 100 peptide binders for each target. Removing hypercone filtering entirely ("w/o filtering") causes a dramatic collapse in half-life, from roughly 30-35 h down to 4-13 h, while leaving non-fouling and solubility largely unchanged, indicating that filtering out poorly aligned moves is essential for optimizing objectives that require gradual, coordinated changes. Introducing static hypercone gating without angle adaptation ("w/o adaptation") recovers much of the half-life gains (to 23-37 h), but at the expense of reduced non-fouling and solubility scores and only marginal improvements in affinity. In contrast, the full MOG-DFM, with both directional hypercone filtering and adaptive angle updates, simultaneously elevates half-life and maintains strong performance across all five objectives. This effect is especially pronounced on disordered targets (4E-BP2 and EWS::FLI1), where dynamic cone adjustment is essential for navigating the irregular, non-convex Pareto landscapes.

## G  ADDITIONAL PROOF

**Claim:** MOG-DFM directs the discrete generation process toward the Pareto front by inducing a positive expected improvement in the direction of a specified weight vector $\omega \in \mathbb{R}^N$.

**Proof:** Let $\mathcal{S} = \mathcal{T}^d$ be the discrete sequence space over vocabulary $\mathcal{T}$, and let $x \in \mathcal{S}$ denote the current sequence state at time $t \in [0, 1]$. Assume the multi-objective score function $s : \mathcal{S} \to \mathbb{R}^N$ is measurable, with $N$ scalar objectives. Define the improvement vector at a candidate transition $y^i \in \mathcal{T} \setminus \{x^i\}$ at position $i \in \{1, \dots, d\}$ as:

$$\Delta s(y^i, x) := s(x^{(i \to y^i)}) - s(x),$$

where $x^{(i \to y^i)}$ denotes the sequence $x$ with token $x^i$ replaced by $y^i$.

Let $\omega \in \mathbb{R}^N$ be a fixed unit-norm trade-off vector sampled uniformly from the Das-Dennis lattice covering the simplex $\Delta^{N-1}$. Define the directional improvement of a transition $y^i$ as:

$$D(y^i, x; \omega) := \Delta s(y^i, x) \cdot \omega.$$

Define the set of feasible transitions (those within the hypercone of angle $\Phi \in (0, \pi)$) at time $t$ as:

$$Y^i(x, \omega, \Phi) := \left\{ y^i \in \mathcal{T} \setminus \{x^i\} \ \middle| \ \arccos\left( \frac{\Delta s(y^i, x) \cdot \omega}{\|\Delta s(y^i, x)\| \cdot \|\omega\|} \right) \leq \Phi \right\}.$$

Let $\mu_t^i(\cdot \mid x, \omega)$ be the conditional probability measure over feasible transitions defined by:

$$\mu_t^i(y^i \mid x, \omega) := \frac{\exp\left( \Delta S(y^i, x, \omega) \right)}{Z(x, \omega)} \cdot \mathbf{1}_{\{y^i \in Y^i(x, \omega, \Phi)\}},$$

where $\Delta S(\cdot)$ is the rank-directional guidance score and $Z(x, \omega) := \sum_{y^i \in Y^i} \exp\left( \Delta S(y^i, x, \omega) \right)$ is the normalizing partition function. Assume that $Y^i(x, \omega, \Phi)$ is non-empty, or else the algorithm falls back to selecting the best $y^i$ with $D(y^i, x; \omega) > 0$ by construction.

We now consider the expected improvement in the direction of $\omega$ over all guided transitions:

$$\mathbb{E}_{i \sim \mathcal{U}[d], \ y^i \sim \mu_t^i(\cdot \mid x, \omega)} \left[ D(y^i, x; \omega) \right] = \frac{1}{d} \sum_{i=1}^{d} \sum_{y^i \in Y^i(x, \omega, \Phi)} D(y^i, x; \omega) \cdot \mu_t^i(y^i \mid x, \omega).$$

Since each $y^i \in Y^i(x, \omega, \Phi)$ satisfies $\arccos\left( \frac{\Delta s(y^i, x) \cdot \omega}{\|\Delta s(y^i, x)\| \cdot \|\omega\|} \right) \leq \Phi < \pi$, it follows that $D(y^i, x; \omega) > 0$ for all $y^i \in Y^i$. Moreover, $\mu_t^i(y^i \mid x, \omega) > 0$ by construction.

Therefore, each term in the sum is strictly positive, and thus:

$$\mathbb{E}[\Delta s(x_{\text{new}}, x) \cdot \omega] > 0,$$

where $x_{\text{new}} = x^{(i \to y^i)}$ is the updated sequence following a guided and filtered transition.

Hence, the MOG-DFM procedure ensures that in expectation, the sampling dynamics induce forward motion along the Pareto trade-off direction $\omega$, thereby steering generation toward the Pareto frontier.

$\square$

# H  ADDITIONAL FIGURES AND TABLES

Table 5: Average solubility (LogS) and half-life (in hours) metrics computed by ADMET-AI for each target across the 100 MOG-DFM-designed binders.

| Target | LogS | Half-Life |
|---|---|---|
| AMHR2 | -2.3931 | 15.505 |
| AMHR2 | -2.5055 | 18.777 |
| AMHR2 | -2.5784 | 16.463 |
| EWS::FLI1 | -2.3869 | 18.945 |
| EWS::FLI1 | -2.3813 | 16.305 |
| EWS::FLI1 | -2.5457 | 15.984 |
| MYC | -2.4053 | 16.491 |
| OX1R | -2.4772 | 23.002 |
| DUSP12 | -2.4333 | 19.258 |
| 1B8Q | -2.3203 | 18.7862 |
| 1E6I | -2.0394 | 19.9358 |
| 3IDJ | -2.4193 | 20.3586 |
| 5AZ8 | -2.5964 | 16.3016 |
| 7JVS | -2.4824 | 20.2565 |

Table 6: **Ablation study results for the adaptive hypercone filtering module in MOG-DFM.** Three settings are evaluated: 'w/o filtering' indicates the module is completely disabled, 'w/o adaptation' means the module is enabled but the hypercone is not adaptive, and 'MOG-DFM' represents the complete algorithm. For each setting, 100 peptide binders were designed, with lengths of 7, 12, and 12 for the targets 3IDJ, 4E-BP2, and EWS::FLI1, respectively.

| Target | Method | Hemolysis ($\downarrow$) | Non-Fouling | Solubility | Half-Life | Affinity |
|---|---|---|---|---|---|---|
| | w/o filtering | 0.0660 | 0.8430 | 0.8482 | 12.50 | 7.3730 |
| 3IDJ | w/o adaptation | 0.0856 | 0.8060 | 0.7970 | 37.17 | 7.3142 |
| | MOG-DFM | 0.0924 | 0.8246 | 0.7992 | 30.39 | 7.6304 |
| | w/o filtering | 0.0504 | 0.8582 | 0.8600 | 12.62 | 6.5066 |
| 4E-BP2 | w/o adaptation | 0.0638 | 0.8418 | 0.8234 | 23.44 | 6.4548 |
| | MOG-DFM | 0.0698 | 0.8210 | 0.8050 | 34.88 | 6.5824 |
| | w/o filtering | 0.0450 | 0.8596 | 0.8570 | 4.40 | 6.1392 |
| EWS::FLI1 | w/o adaptation | 0.0620 | 0.8444 | 0.8482 | 28.82 | 6.2118 |
| | MOG-DFM | 0.0616 | 0.8302 | 0.8130 | 34.225 | 6.3631 |

Table 7: Ablation results for peptide binder design targeting PDB 7LUL with different guidance settings. For each setting, 100 binders of length 7 were designed.

| Guidance Settings | | | Affinity | Solubility | Hemolysis ($\downarrow$) |
|---|---|---|---|---|---|
| Affinity | Solubility | Hemolysis | | | |
| ✓ | ✓ | ✓ | 6.3489 | 0.8890 | 0.0620 |
| × | ✓ | ✓ | 5.0514 | 0.9482 | 0.0406 |
| ✓ | × | ✓ | 6.9060 | 0.4224 | 0.0488 |
| ✓ | ✓ | × | 6.5304 | 0.8975 | 0.1019 |
| × | × | ✓ | 5.0761 | 0.7148 | 0.0163 |
| × | ✓ | × | 5.2434 | 0.9772 | 0.0955 |
| ✓ | × | × | 7.4834 | 0.1218 | 0.3281 |
| × | × | × | 5.5631 | 0.3736 | 0.1567 |

Table 8: Ablation results for peptide binder design targeting PDB CLK1 with different guidance settings. For each setting, 100 binders of length 12 were designed.

| Guidance Settings | | | Affinity | Non-Fouling | Half-Life |
|---|---|---|---|---|---|
| Affinity | Non-Fouling | Half-Life | | | |
| ✓ | ✓ | ✓ | 6.9194 | 0.7401 | 51.73 |
| × | ✓ | ✓ | 6.4735 | 0.8107 | 60.75 |
| ✓ | × | ✓ | 7.5360 | 0.3062 | 84.70 |
| ✓ | ✓ | × | 7.4150 | 0.8560 | 1.24 |
| × | × | ✓ | 6.2363 | 0.2624 | 96.44 |
| × | ✓ | × | 6.1378 | 0.9503 | 0.94 |
| ✓ | × | × | 8.5943 | 0.2439 | 3.15 |
| × | × | × | 5.8926 | 0.3999 | 1.94 |

Table 9: **Performance evaluation of MOG-DFM in guided DNA sequence generation.** Task 1 guides the generation towards the HelT shape and enhancer class 1, while Task 2 targets the Rise shape and enhancer class 16. The table presents the predicted DNA shape values (HelT for Task 1, Rise for Task 2) and enhancer class probabilities (class 1 for Task 1, class 16 for Task 2) under various guidance conditions. The 'Shape' column shows the predicted DNA shape values obtained using Deep DNAshape, and the 'Class Prob' column displays the predicted enhancer class probabilities. Ablation studies were conducted by removing one or both guidance criteria, as shown by the rows corresponding to different combinations of shape and class guidance. For each setting, 5 enhancer DNA sequences were designed.

| Guidance Settings | | Task 1 | | Task 2 | |
|---|---|---|---|---|---|
| **Shape** | **Class** | **Class Prob** | **Shape** | **Class Prob** | **Shape** |
| | | 0.7504 | 36.0100 | 0.9960 | 3.3640 |
| | | 0.6507 | 36.0100 | 0.9922 | 3.3680 |
| ✓ | ✓ | 0.6821 | 36.0000 | 0.9864 | 3.3669 |
| | | 0.7097 | 36.0000 | 0.9976 | 3.3680 |
| | | 0.6425 | 36.0000 | 0.9961 | 3.3623 |
| | | 0.9999 | 34.3274 | 1.0000 | 3.3368 |
| | | 0.9999 | 34.4715 | 1.0000 | 3.3345 |
| ✓ | ✗ | 0.9989 | 34.4257 | 0.9999 | 3.3348 |
| | | 0.9997 | 34.5226 | 0.9994 | 3.3357 |
| | | 0.9998 | 34.4210 | 1.0000 | 3.3340 |
| | | 0.0026 | 36.0017 | 2.36E-05 | 3.3690 |
| | | 0.0055 | 36.0238 | 0.0005 | 3.3647 |
| ✗ | ✓ | 0.0062 | 36.0214 | 0.0114 | 3.3705 |
| | | 0.0186 | 36.0396 | 0.0001 | 3.3717 |
| | | 0.0051 | 36.0304 | 0.0054 | 3.3669 |
| | | 0.0362 | 34.7379 | 0.0008 | 3.3283 |
| | | 0.0364 | 34.5350 | 0.0057 | 3.3258 |
| ✗ | ✗ | 0.0309 | 34.5720 | 0.0476 | 3.3268 |
| | | 0.0138 | 34.3060 | 0.0632 | 3.3378 |
| | | 0.0213 | 34.5500 | 0.0003 | 3.3320 |

Table 10: Hyperparameter sensitivity benchmark for MOG-DFM in peptide binder generation, guided by five objectives. For each setting, 100 peptide binders are designed with a length matching that of the pre-existing binder for each target.

| Hyper parameter | Target | Value | Hemolysis ($\downarrow$) | Non-Fouling | Solubility | Half-Life | Affinity |
|---|---|---|---|---|---|---|---|
| num_div | 6MLC | 32 | 0.0994 | 0.8088 | 0.7924 | 38.39 | 6.5436 |
| | | 64 | 0.0863 | 0.8280 | 0.8232 | 34.91 | 6.3260 |
| | | 128 | 0.0890 | 0.8438 | 0.8386 | 32.97 | 6.4197 |
| $\beta$ | 4IU7 | 0.5 | 0.0829 | 0.7894 | 0.761 | 28.10 | 6.7884 |
| | | 1 | 0.0684 | 0.8388 | 0.8321 | 41.78 | 7.0002 |
| | | 1.5 | 0.0585 | 0.8588 | 0.8582 | 47.65 | 7.0505 |
| | | 2 | 0.0615 | 0.8461 | 0.8416 | 53.45 | 7.0169 |
| $\lambda$ | 1AYC | 0.5 | 0.0703 | 0.8168 | 0.8152 | 30.89 | 6.4838 |
| | | 1 | 0.0647 | 0.8362 | 0.8207 | 33.28 | 6.4549 |
| | | 2 | 0.0587 | 0.8690 | 0.8461 | 41.90 | 6.5317 |
| $\alpha_r$ | 2Q8Y | 0.1 | 0.0777 | 0.8361 | 0.8051 | 37.83 | 6.0569 |
| | | 0.3 | 0.0718 | 0.8441 | 0.8280 | 38.83 | 6.0484 |
| | | 0.5 | 0.0718 | 0.8529 | 0.8421 | 31.45 | 6.0445 |
| | | 0.7 | 0.0688 | 0.8403 | 0.8377 | 35.50 | 6.0839 |
| | | 0.9 | 0.0813 | 0.8288 | 0.8091 | 45.25 | 6.1599 |
| $\eta$ | 2LTV | 0.5 | 0.0633 | 0.8437 | 0.8368 | 29.48 | 7.3657 |
| | | 1 | 0.0601 | 0.8256 | 0.8144 | 24.47 | 7.3111 |
| | | 2 | 0.0624 | 0.8125 | 0.7887 | 35.13 | 7.1974 |
| $\Phi_{init}$ | 5M02 | 15 | 0.0746 | 0.8285 | 0.8007 | 34.04 | 7.0335 |
| | | 30 | 0.0792 | 0.8393 | 0.8187 | 35.60 | 7.0251 |
| | | 45 | 0.0747 | 0.8338 | 0.8192 | 36.29 | 7.0944 |
| | | 60 | 0.0813 | 0.8095 | 0.7970 | 38.25 | 7.0932 |
| | | 75 | 0.0830 | 0.8139 | 0.7949 | 33.29 | 7.1261 |
| $[\Phi_{min}, \Phi_{max}]$ | 3EQS | [0,90] | 0.0572 | 0.8385 | 0.8200 | 26.64 | 8.2201 |
| | | [15,75] | 0.0599 | 0.8373 | 0.8116 | 29.56 | 8.1673 |
| | | [30,60] | 0.0614 | 0.8159 | 0.8020 | 35.71 | 8.2313 |
| $\tau$ | 5E1C | 0 | 0.0614 | 0.8252 | 0.8119 | 24.57 | 7.0112 |
| | | 0.1 | 0.0650 | 0.8017 | 0.7835 | 31.19 | 7.1067 |
| | | 0.3 | 0.0595 | 0.8224 | 0.8088 | 28.72 | 7.0756 |
| | | 0.5 | 0.0555 | 0.8310 | 0.8043 | 24.03 | 7.0862 |
| | | 0.7 | 0.0590 | 0.8360 | 0.8078 | 28.27 | 7.0477 |
| $T$ | 5KRI | 50 | 0.0757 | 0.7386 | 0.7219 | 15.22 | 6.9155 |
| | | 100 | 0.0580 | 0.8617 | 0.8504 | 30.25 | 6.9946 |
| | | 200 | 0.0525 | 0.8695 | 0.8621 | 41.53 | 7.2166 |
| | | 500 | 0.0518 | 0.8799 | 0.8760 | 57.65 | 7.2172 |

Table 11: Motif clusters and associated properties of enhancer DNA sequences. In this paper, each class refers to its corresponding cluster ID.

| Cluster ID | # of explainable ASCAVs | Motif Annotation | # of Motifs in the cluster |
|---|---|---|---|
| cluster_1 | 3278 | ATF | 71 |
| cluster_2 | 1041 | CTCF | 85 |
| cluster_3 | 2480 | EBOX | 91 |
| cluster_4 | 4011 | AP1 | 191 |
| cluster_5 | 1165 | RUNX | 37 |
| cluster_6 | 789 | SP | 20 |
| cluster_7 | 1285 | ETS | 33 |
| cluster_8 | 544 | TEAD | 9 |
| cluster_9 | 1024 | TFAP | 53 |
| cluster_10 | 334 | Other | 4 |
| cluster_11 | 935 | SOX | 17 |
| cluster_12 | 1010 | CTCFL | 16 |
| cluster_13 | 696 | GATA | 7 |
| cluster_14 | 141 | Other | 2 |
| cluster_15 | 601 | TEAD | 6 |
| cluster_16 | 805 | Other | 7 |
| cluster_17 | 270 | Other | 4 |
| cluster_18 | 475 | Other | 5 |
| cluster_19 | 473 | ZNF | 6 |
| cluster_20 | 395 | Other | 4 |
| cluster_21 | 393 | Other | 4 |
| cluster_22 | 768 | NRF | 8 |
| cluster_23 | 214 | Other | 2 |
| cluster_24 | 336 | Other | 2 |
| cluster_25 | 375 | Other | 3 |
| cluster_26 | 215 | Other | 2 |
| cluster_27 | 234 | Other | 2 |
| cluster_28 | 354 | Other | 3 |
| cluster_29 | 210 | Other | 2 |
| cluster_30 | 200 | Other | 2 |
| cluster_31 | 218 | Other | 2 |
| cluster_32 | 415 | Other | 2 |
| cluster_33 | 387 | SOX | 2 |
| cluster_34 | 116 | Other | 1 |
| cluster_35 | 121 | Other | 1 |
| cluster_36 | 394 | Other | 2 |
| cluster_37 | 112 | Other | 1 |
| cluster_38 | 111 | Other | 1 |
| cluster_39 | 107 | Other | 1 |
| cluster_40 | 118 | Other | 1 |
| cluster_41 | 144 | Other | 1 |
| cluster_42 | 105 | Other | 1 |
| cluster_43 | 102 | Other | 1 |
| cluster_44 | 108 | Other | 1 |
| cluster_45 | 114 | Other | 1 |
| cluster_46 | 118 | Other | 1 |
| cluster_47 | 119 | Other | 1 |

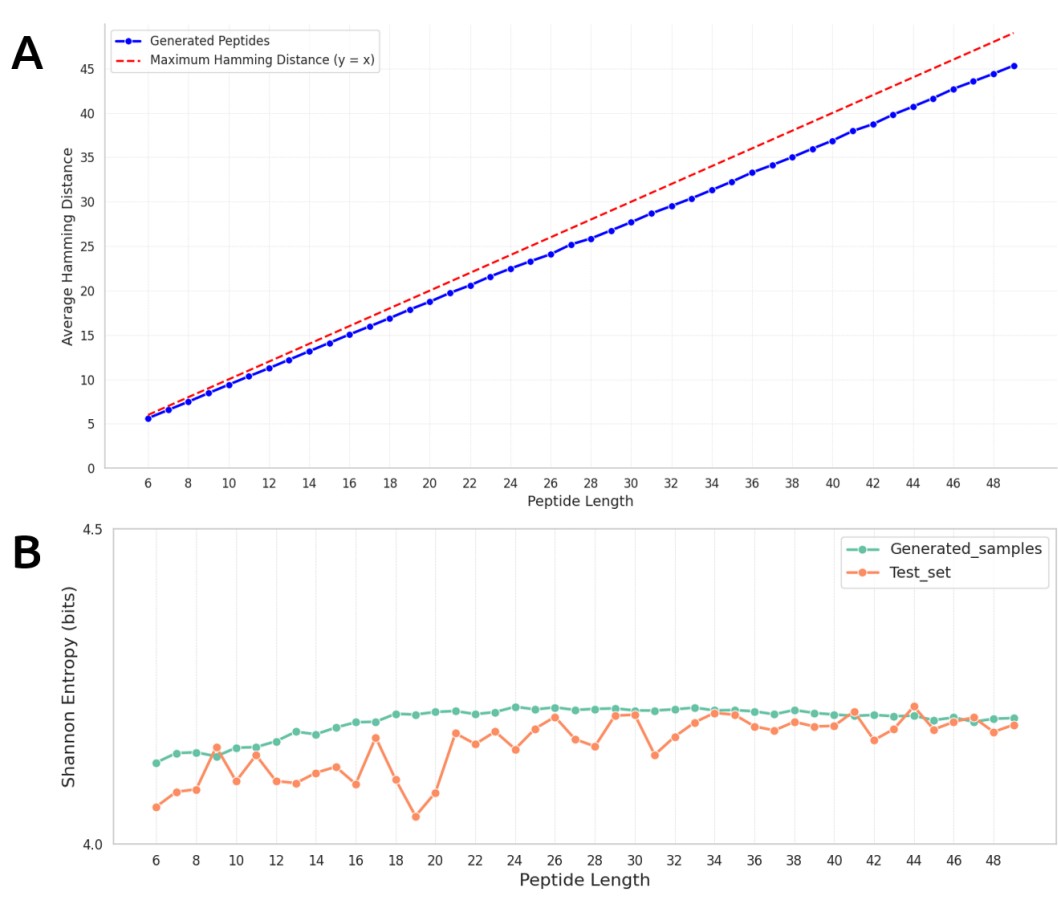

Figure A1: **(A)** The Hamming distance of sampled peptides of different lengths to the peptides of the same length in the test set. **(B)** The Shannon Entropy of sampled peptides of different lengths to the peptides of the same length in the test set.

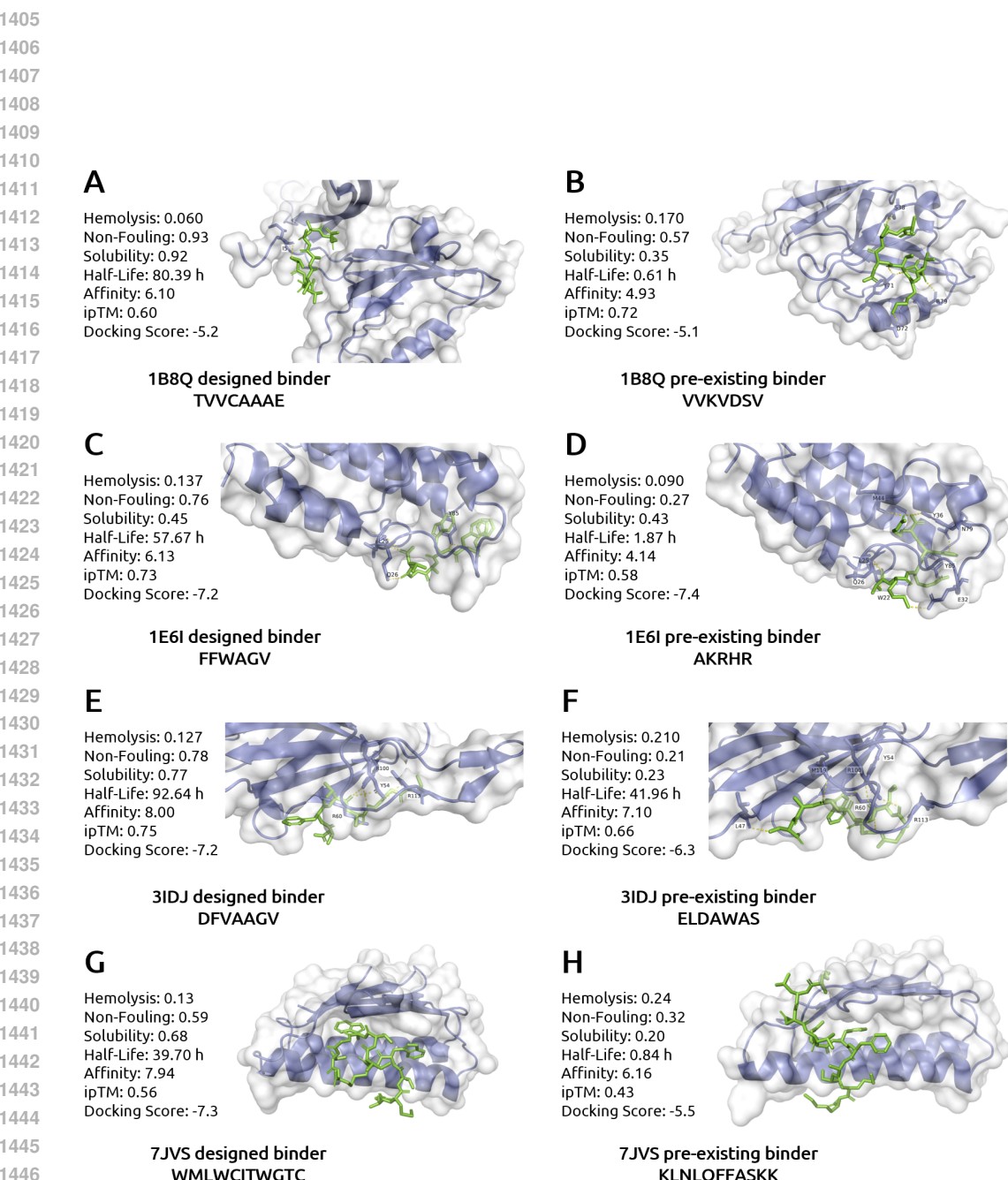

Figure A2: **Complex structures of target proteins with pre-existing binders.** **(A)-(B)** 1B8Q, **(C)-(D)** 1E6I, **(E)-(F)** 3IDJ, **(G)-(H)** 7JVS. Each panel shows the complex structure of the target with either a MOG-DFM-designed binder or its pre-existing binder. For each binder, five property scores are provided, as well as the ipTM score from AlphaFold3 and the docking score from AutoDock VINA. Interacting residues on the target are visualized.

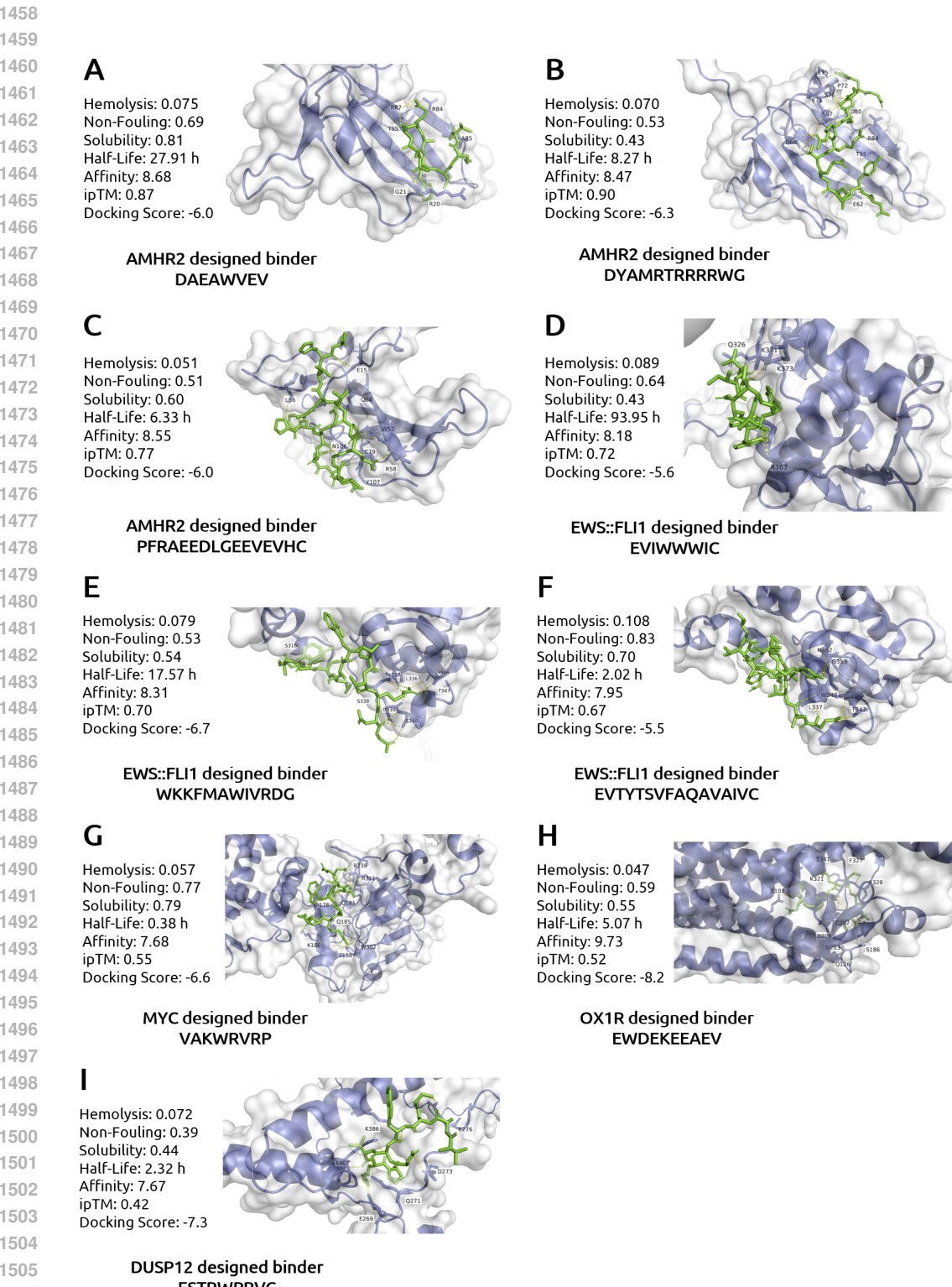

**A**

Hemolysis: 0.075
Non-Fouling: 0.69
Solubility: 0.81
Half-Life: 27.91 h
Affinity: 8.68
ipTM: 0.87
Docking Score: -6.0

**AMHR2 designed binder
DAEAWVEV**

**B**

Hemolysis: 0.070
Non-Fouling: 0.53
Solubility: 0.43
Half-Life: 8.27 h
Affinity: 8.47
ipTM: 0.90
Docking Score: -6.3

**AMHR2 designed binder
DYAMRTRRRRWG**

**C**

Hemolysis: 0.051
Non-Fouling: 0.51
Solubility: 0.60
Half-Life: 6.33 h
Affinity: 8.55
ipTM: 0.77
Docking Score: -6.0

**AMHR2 designed binder
PFRAEEDLGEEVEVHC**

**D**

Hemolysis: 0.089
Non-Fouling: 0.64
Solubility: 0.43
Half-Life: 93.95 h
Affinity: 8.18
ipTM: 0.72
Docking Score: -5.6

**EWS::FLI1 designed binder
EVIWWWIC**

**E**

Hemolysis: 0.079
Non-Fouling: 0.53
Solubility: 0.54
Half-Life: 17.57 h
Affinity: 8.31
ipTM: 0.70
Docking Score: -6.7

**EWS::FLI1 designed binder
WKKFMAWIVRDG**

**F**

Hemolysis: 0.108
Non-Fouling: 0.83
Solubility: 0.70
Half-Life: 2.02 h
Affinity: 7.95
ipTM: 0.67
Docking Score: -5.5

**EWS::FLI1 designed binder
EVTYTSVFAQAVAIVC**

**G**

Hemolysis: 0.057
Non-Fouling: 0.77
Solubility: 0.79
Half-Life: 0.38 h
Affinity: 7.68
ipTM: 0.55
Docking Score: -6.6

**MYC designed binder
VAKWRVRP**

**H**

Hemolysis: 0.047
Non-Fouling: 0.59
Solubility: 0.55
Half-Life: 5.07 h
Affinity: 9.73
ipTM: 0.52
Docking Score: -8.2

**OX1R designed binder
EWDEKEEAEV**

**I**

Hemolysis: 0.072
Non-Fouling: 0.39
Solubility: 0.44
Half-Life: 2.32 h
Affinity: 7.67
ipTM: 0.42
Docking Score: -7.3

**DUSP12 designed binder
ESTRWPRVC**

Figure A3: **Complex structures of target proteins without pre-existing binders. (A)-(C)** AMHR2, **(D)-(F)** EWS::FLI1, **(G)** MYC, **(H)** OX1R, **(I)** DUSP12. Each panel shows the complex structure of the target with a MOG-DFM-designed binder. For each binder, five property scores are provided, as well as the ipTM score from AlphaFold3 and the docking score from AutoDock VINA. Interacting residues on the target are visualized.

---

**Algorithm 1** MOG-DFM: Multi-Objective-Guided Discrete Flow Matching

---

1: **Input:** Pre-trained discrete flow matching model, multi-objective score functions
2: **Output:** Sequence $x_1$ with multi-objective optimized properties
3: **Initialize:**
4:     Sample an initial sequence $x_0$ uniformly from the discrete state space $S$
5:     Generate a set of weight vectors $\{\omega_k\}_{k=1}^M$ that uniformly cover the N-dimensional Pareto front
6:     Select a weight vector $\omega$ randomly from $\{\omega_k\}$
7: **for** $t = 0$ to $1$ with step size $h = \frac{1}{T}$ **do**
8:     **Step 1: Guided Transition Scoring**
9:       Select a position $i$ in the sequence to update
10:       For each candidate transition $y_i \neq x_i$:
11:         Compute the normalized rank score $I_n(y_i, x)$ for each objective $n$
12:         Compute $D(y_i, x, \omega)$ based on the alignment of improvements with $\omega$
13:         Combine rank and direction components:

$$\Delta S(y_i, x, \omega) = \text{Norm} \left[ \frac{1}{N} \sum_{n=1}^N I_n(y_i, x) \right] + \lambda \cdot \text{Norm} \left[ D(y_i, x, \omega) \right]$$

14:       Re-weight the original velocity field $u_i(y_i, x)$ by the combined score
15:     **Step 2: Adaptive Hypercone Filtering**
16:       Compute angle $\alpha_i$ between improvement vector $\Delta s(y_i, x)$ and weight vector $\omega$
17:       Accept transitions $y_i$ where $\alpha_i \leq \Phi$ (hypercone angle)
18:       Select the best transition $y_i^{best}$ from the candidates
19:       **Adapt Hypercone Angle:**
20:         Compute the rejection rate $r_t$ based on the number of rejected candidate transitions
21:         Compute the exponential moving average $\overline{r_t}$ of rejection rate
22:         Update the hypercone angle $\Phi$ based on the moving average:

$$\Phi_{t+h} = \text{clip} \left( \Phi_t \exp \left( \eta \left( \overline{r_t} - \tau \right) \right), \Phi_{\min}, \Phi_{\max} \right)$$

23:     **Step 3: Euler Sampling**
24:       Use Euler's method to sample the next state based on the guided velocity field
25:       Transition to the new sequence
26:       Update time: $t \rightarrow t + h$
27: **end for**
28: **Return:** Final sequence $x_1$

---

# I USE OF LARGE LANGUAGE MODELS (LLMS)

We acknowledge the use of large language models (LLMs) to assist in polishing and editing parts of this manuscript. LLMs were used to refine phrasing, improve clarity, and ensure consistency of style across sections. All technical content, experiments, analyses, and conclusions were developed by the authors, with LLM support limited to language refinement and editorial improvements.

