# OpenReview forum: "Multi-Objective-Guided Discrete Flow Matching for Controllable Biological Sequence Design"
_ICLR.cc/2026/Conference — ICLR 2026 Conference Withdrawn Submission_

### Official Review · Reviewer_uKWQ · 2025-10-24

**Soundness:** 2
**Presentation:** 2
**Contribution:** 3
**Rating:** 6
**Confidence:** 3

**Summary:**

The paper proposes MOG-DFM (Multi-Objective-Guided Discrete Flow Matching), a framework to steer pretrained discrete flow matching (DFM) generators toward Pareto-efficient trade-offs over multiple scalar objectives for biological sequence design. The authors train two unconditional DFM bases—PepDFM for peptides and EnhancerDFM for enhancer DNA. Empirically, MOG-DFM improves multi-property profiles over evolutionary baselines.

**Strengths:**

1. Clear problem formulation and practical relevance. Multi-objective sequence design with discrete tokens is important in therapeutic peptide and regulatory DNA design. Framing controllability directly over discrete token transitions addresses a real gap in methods that assume continuous embeddings.

2. Clarity and structure. The paper is generally well written, with a clean decomposition (score, direction, filtering).

3. The method seems to have solid empirical evidence on two modalities. Any wet-lab evaluations?

**Weaknesses:**

1. Validation relies heavily on proxy predictors with limited external ground truth. It seems the oracle predictor are not reliable. Some paper https://arxiv.org/abs/2503.17286 discussed some lookup-table oracle/task where you may want to use, or at least discuss.

2. While tables report per-objective means, there is no traditional metrics like hypervolume (HV).

3. There seems a lack of error bars/variances in the paper.

4. The rank-directional score is overlly complicated and needs to tune many hyperparameters manually. I do not think it is scalable and might overfit to the tasks. Why not just compute the predictor's grad in each position using some discrete-gradient opt techs?

5. What is the hyperparameter setting of Peptune in this context? I think we should scale the MCTS tree depth/width to match the compute used  to make a fair comparison.

**Questions:**

See Weakness.

---

### Official Review · Reviewer_5dzb · 2025-10-25

**Soundness:** 2
**Presentation:** 2
**Contribution:** 2
**Rating:** 2
**Confidence:** 3

**Summary:**

This paper introduces MOG-DFM, a framework for performing multi-objective optimization in discrete flow matching. The method proposes two main components:
(1) a rank–directional score (ΔS) combining rank-based improvement and alignment with sampled Pareto trade-off directions, and
(2) an adaptive hypercone filter that constrains sampling to Pareto-consistent regions.
The paper claims to achieve Pareto-consistent generation and better trade-off control without retraining, evaluated on biological sequence design tasks.

**Strengths:**

The paper introduces a new paradigm for performing multi-objective optimization directly within a discrete flow-matching model. The hypercone filtering mechanism is novel and the rank–directional combination is original and intuitive, offering a practical way to handle non-differentiable or noisy objectives.

Applies the method to two distinct domains (DNA and peptides) and compares against multiple baselines (diffusion, evolutionary, and hybrid methods).

**Weaknesses:**

Many of its design features lack a more intuitive explanation and stronger motivation, such as exponential reweighting and the use of EMA to maintain $\Phi$. Subsequent ablation experiments reinforce this concern. For example, in Table 6, w/o filtering and w/o adaptation achieve better performance on the Hemolysis metric.
The paper would benefit from consistent variable definitions and a clearer flow between subsections. Some mathematical symbols, e.g. $\mathcal{T},T,[K]^d$, should be clearly explained and formally defined. The rank function does not have a specific form, and readers are not sure how to implement it.

**Questions:**

See weakness. Should $\pi$ in section 2.1.3 be $\frac{\pi}{2}$?

---

### Official Review · Reviewer_ov8v · 2025-11-01

**Soundness:** 2
**Presentation:** 2
**Contribution:** 1
**Rating:** 2
**Confidence:** 4

**Summary:**

The authors propose MOG-DFM, a multi-objective-guided discrete flow matching method to steer the discrete flow matching generative models towards Pareto-efficient trade-offs across multiple objectives. Experiments on peptide generation and enhancer generation showcase the effectiveness of MOG-DFM.

**Strengths:**

- The problem statement and the proposed method are well-motivated.
- MOG-DFM extends discrete flow matching to support Pareto-guided generation across multiple objectives.

**Weaknesses:**

- The proposed method seems to be a simple adaptation of the ParetoFlow method [1] to the discrete sequence case, where each of the key steps shares significant similarity to ParetoFlow. This limits the novelty and methodological contribution of the proposed method. Also, the difference and novelty, especially in comparison to ParetoFlow are not clearly stated.
- In step 1, the authors randomly select one position on the sequence to update. This is very inefficient, especially when the sequence length is long.
- In Table 3, MOG-DFM takes a significantly longer time than PepTune+DPLM. Therefore, a naive improvement of PepTune+DPLM under the same computational cost would be generating multiple samples from PepTune+DPLM (to keep the total time the same as MOG-DFM) and utilizing best-of-N on top of these samples. This should offer a fair comparison between PepTune+DPLM and MOG-DFM.
- The experimental results lack comparison to baseline methods. All the tables and figures, except for table 3, only present the performance of MOG-DFM or its comparison with the pretrained model without any guidance. Even in table 3, only PepTune and other very traditional MOO methods are compared with. The comparison with other reward-guided sampling baselines or RL fine-tuning baselines is missing. Although these guidance-based methods are not designed for Pareto optimization, they can be naively adapted to achieve multi-objective optimization by enforcing certain weights to balance these objectives into a single aggregated objective. Also, how does the model perform compared with [2]?



[1] ParetoFlow: Guided Flows in Multi-Objective Optimization. ICLR 2025.

[2] Uncertainty-Aware Multi-Objective Reinforcement Learning-Guided Diffusion Models for 3D De Novo Molecular Design. NeurIPS 2025.

**Questions:**

Please refer to the **Weaknesses** section.

---

### Official Review · Reviewer_3CNY · 2025-11-01

**Soundness:** 2
**Presentation:** 2
**Contribution:** 2
**Rating:** 2
**Confidence:** 3

**Summary:**

This paper studies the problem of multi-objective biological sequence generation. The paper proposes a discrete flow matching approach to achieve this goal. Experiments are conducted on multi-objective peptide binder generation and multi-objective DNA sequence generation.

**Strengths:**

- The problem of multi-objective sequence generation is an important problem. Specifically, generating sequences that bind strongly to target molecules while exhibiting low affinity for off-targets has significant practical value.
- Performing biological sequence generation in a discrete space offers advantages and strengths.

**Weaknesses:**

- The main weakness of this paper is the lack of technical motivation for the proposed method. The method consists of four steps for sequence generation. However, the paper does not explain why these specific steps are expected to improve the generation process. There are several simpler approaches for multi-objective generation. However, the paper does not discuss the advantages of the proposed method relative to these alternatives. This weakens the overall contribution.
- The experimental section could also be strengthened. More baselines should be added and a brief introduction or description of the baselines should be provided in the main text, as this information appears to be missing.

**Questions:**

- I suggest adding an overview of the proposed method and the motivations underlying its novelty at the beginning of Section 2.
- It is also recommended to include a preliminary section in the main text that explains discrete flow matching. Since there are some blank spaces in the paper, the experimental section could be slightly shortened to make room for this preliminary discussion.
- One approach to multi-objective sequence generation is the use of LLMs. Is it correct that a discrete flow matching model requires less data for pre-training compared to an LLM? It would be beneficial for the paper to discuss this point as well.

---

### Note · Authors · 2025-11-20

**Comment:**

Thank you all for the time and care you put into reviewing our submission. We appreciate the detailed feedback and fully agree that addressing the requested experimental work and additional *in silico* comparisons will require significantly more development than can be completed during the current review cycle. We are already in the process of incorporating new experimental wet-lab data to support MOG-DFM's conclusions, and several of the recommended analyses are substantial enough that we would prefer to complete them thoroughly rather than rush partial updates that would not do justice to the reviewers’ suggestions.

Given these constraints, we believe the most respectful course is to withdraw the paper at this stage so as not to take up further reviewer or AC bandwidth. We plan to prepare a more complete version of the work, integrating both experimental validation and expanded baselines, for a journal submission. We are grateful for the constructive feedback, which will materially improve the final manuscript.

**Withdrawal Confirmation:**

I have read and agree with the venue's withdrawal policy on behalf of myself and my co-authors.